# Gender differences in factors influencing intention to undergo cardiovascular disease health checks: A cross-sectional survey

Ai Theng Cheong[1]*, Seng Fah Tong[2], Karuthan Chinna[3], Ee Ming Khoo[4], Su May Liew[4]

1 Department of Family Medicine, Faculty of Medicine and Health Sciences, Universiti Putra Malaysia, Serdang, Selangor, Malaysia, 2 Department of Family Medicine, Faculty of Medicine, Universiti Kebangsaan Malaysia, Kuala Lumpur, Malaysia, 3 School of Medicine, Faculty of Health and Medical Sciences, Taylor's University Malaysia, Subang Jaya, Malaysia, 4 Department of Primary Care Medicine, University of Malaya Primary Care Research Group (UMPCRG), Faculty of Medicine, University of Malaya, Kuala Lumpur, Malaysia

* cheaitheng@upm.edu.my, caitheng@gmail.com

## Abstract

### Background

Undergo a health check for cardiovascular disease (CVD) is an important strategy to improve cardiovascular (CV) health. Men are reported to be less likely to undergo cardiovascular disease (CVD) health check than women. Gender difference could be one of the factors influencing health seeking behaviour of men and women. We aimed to identify gender differences in factors influencing the intention to undergo CVD health checks.

### Methods

This was a cross-sectional survey using mall intercept interviews. Malaysians aged ≥30 years without known CVD were recruited. They were asked for their intention to undergo CVD health checks and associated factors. The factors included seven internal factors that were related to individuals' attitude, perception and preparedness for CVD health checks and two external factors that were related to external resources. Hierarchical ordinal regression analysis was used to evaluate the importance of the factors on intention to undergo CVD health checks, for men and women separately.

### Results

397 participants were recruited, 60% were women. For men, internal factors explained 31.6% of the variances in likeliness and 9.6% of the timeline to undergo CVD health checks, with 1.2% and 1.8% added respectively when external factors were sequentially included. For women, internal factors explained 18.9% and 22.1% of the variances, with 3.1% and 4.2% added with inclusion of the external factors. In men, perceived drawbacks of health checks was a significant negative factor associated with likeliness to undergo CVD health checks (coefficient = -1.093; 95%CI:-1.592 to -0.594), and timeline for checks (coefficient = -0.533; 95%CI:-0.975 to -0.091). In women, readiness to handle outcomes following health

**Data Availability Statement:** All relevant data are within the manuscript and its Supporting Information files.

**Funding:** Financial support of this study was provided by University of Malaya Postgraduate Research Fund (PG049-2014A) via EMK. The funder have no role in the study design, data collection and analysis, decision to publish, or preparation of the manuscript. (URL of funder website: https://umresearch.um.edu.my/funding-opportunities-information).

**Competing interests:** The authors have declared that no competing interests exist.

checks was significantly associated with likeliness to undergo the checks (coefficient = 0.575; 95%CI: 0.063 to 1.087), and timeline for checks (coefficient = 0.645; 95%CI: 0.162 to 1.128). Both external factors 1) influence by significant others (coefficient = 0.406; 95%CI: 0.013 to 0.800) and 2) external barriers (coefficient = -0.440; 95%CI:-0.869 to -0.011) were also significantly associated with likeliness to undergo CVD health checks in women.

## Conclusions

Both men and women were influenced by internal factors in their intention to undergo CVD health checks, and women were also influenced by external factors. Interventions to encourage CVD health checks need to focus on internal factors and be gender sensitive.

## Introduction

Cardiovascular disease (CVD) is a major health burden worldwide including the low- and middle-income countries such as Malaysia [1]. The Lancet Non-Communicable Disease (NCD) Action Group and the NCD Alliance, had proposed to reduce cardiovascular risk among those identified at high risk as a priority to combat non communicable diseases [2]. In most countries, including the low- and middle-income countries, pharmacological treatment for high-risk individuals has been shown to be cost-effective and affordable [3,4].

To enable timely treatment and risk reduction in CVD, identifying individuals at high risk through health checks for disease risk factors in primary care is important [5]. This is especially so for low- and middle-income countries where the prevalence of cardiovascular risk factors are high and awareness was poor [6]. Studies from Korea and Japan have reported that health screening for CVD was associated with lower rates of CVD, all-cause mortality, CVD events, healthcare utilization and costs [7,8]. However, total cardiovascular risk assessment for patients at high-risk has not been optimal in primary care. The uptake rates among high-risk patients invited for health checks ranged between 30% and 40% in developed countries, for example the NHS health checks in England and cardiometabolic risk factor screening in the Netherlands [9–12].

In Malaysia, the prevalence of cardiovascular risk factors is high. The National Health Morbidity Survey in 2015 had reported that 30.3% of the adult population had hypertension, 17.5% had diabetes and 47.7% had hypercholesterolaemia, and more than half of these cases were undiagnosed [13]. In addition, the uptake of health checks for CVD risk factors remains low in Malaysia, ranging from 20% to 40% [14,15].

Women's health check uptake rate was higher than men [10,14,16,17]; 49.7% vs 38.3% respectively in the UK NHS data [10]. Similarly in Malaysia, the health check uptake rate was 40.7% in women vs 34.9% in men in the National Health Morbidity Survey and 24% vs 16% respectively in the Malaysia Social Security Organisation (SOCSO) Health Screening Programme (HSP) [14,16]. This variation in health check uptake is often attributed to differences in the health seeking behaviours of men and women. Men generally engage less with health care compared to women because it fits the traditional masculine image [18]. Men would engage in health promotion only when it is deemed necessary [19]. Hence, identifying factors that influence men's and women's intention to undergo CVD health checks is important to help design gender sensitive and effective strategies to promote health checks participation. Therefore, this study aimed to identify factors that influenced the intention to undergo CVD heath check, and to compare differences among men and women.

## Materials and methods

This was a sub-analysis of a larger study that examined factors influencing individuals' intention to undergo CVD health checks. The detailed method of this study was published elsewhere [20] and is summarized in brief here.

### Study design and data collection

This was a cross sectional survey using mall intercept interviews [21]. Malaysians aged ≥30 years who attended a hypermarket were invited to participate in the survey using convenient sampling. Individuals without known CVD and could understand the Malay language (Malaysia's national language) were recruited. Each Participant completed a self-administered validated CVD health check questionnaire [22]. (Refer S1 File). In circumstances where the participants needed assistance, the researcher would read the questionnaire aloud to them.

The validated questionnaire was used to assess factors that influenced individual's intention to undergo CVD health checks. The questionnaire has 9 factors and 36 items with good factor loading (more than 0.40) and internal consistency (Cronbach's alpha ranged: 0.66–0.85) [22]. Seven were internal factors related to individuals' attitude, perception and preparedness for CVD health checks and two were external factors related to external resources [23].

The seven internal factors were: 1) individuals' belief that the course of CVD can be changed for the better, 2) perceptions of self being at risk of CVD, 3) perception of benefits of CVD health checks, 4) perception of drawbacks of CVD health checks, 5) preferred method for CVD prevention, 6) individuals' readiness to know the results of CVD health checks, 7) individuals' readiness to handle outcomes following CVD health checks and the two external factors were: 1) external barriers, which included financial and time constraints, accessibility to health checks, and 2) influence by significant others. Each item was scored with a Likert scale of 1 to 5; 1 indicated "strongly disagree" and 5 indicated "strongly agree".

To assess participant's intention to undergo CVD health checks, participants were asked about their likeliness to undergo health checks in a specified timeline (within 3 months, 6 months or 1 year). A Likert scale (1: "very unlikely" to 5: "very likely") was used to denote the intention to undergo CVD health checks.

For the regression model, the sample size needed was calculated based on the work of Peduzzi et al [23] using the following formula:

$$N = \frac{10k}{P}$$

Where k was the number of independent variables (9 factors were examined in this study) and P was the proportions of intention to undergo CVD health check. The P was estimated from the intention to undergo for CVD health check reported as 83.7% [20]. Thus, the estimated sample size required was 108 for men and women respectively. The larger study had recruited 397 participants, providing adequate power for this analysis.

### Data analysis

The participant's intention to undergo CVD health checks was assessed by two variables; 1) the degree of likeliness to undergo health checks and 2) the likely timeline to undergo health checks. The degree of likeliness a participant would undergo a health check was measured on a five-point scale; 1(very unlikely), 2 (unlikely), 3 (not sure), 4 (likely) and 5 (very likely). The likely timeline to undergo health checks was measured by a four-point scale: 1 (not sure or not likely to attend), 2 (likely to attend within one year), 3 (likely to attend within 6 months) and 4 (likely to attend within 3 months).

The independent variables were the seven internal factors (individuals' belief that the course of CVD can be changed for the better, perceptions of self being at risk of CVD, perception of benefits of CVD health checks, perception of drawbacks of CVD health checks, preferred method for CVD prevention, individuals' readiness to know the results of CVD health checks, individuals' readiness to handle outcomes following CVD health checks) and the two external factors (external barriers, influence by significant others) of the intention to undergo CVD health checks. A positive relationship was expected between all the factors and the outcome variables except for three i.e. "perceived drawbacks of CVD health checks", "preferred method for CVD prevention (healthy practice vs medical measures)" and "external barriers towards CVD health checks", where a negative relationship was expected.

The hierarchical ordinal regression analysis was used to evaluate the relative importance of the internal and external factors on participants' intention to undergo CVD health checks, for men and women separately. This hierarchical ordinal regression was chosen because we would like to know in addition to the internal factors, the effect of external factors on participants' intention to undergo CVD health checks. We hypothesized the internal factors would precede the external factors when participants intended to undergo health checks. Hence, the internal factors were included in the first block of the regression followed by the external factors. The demographics (age group, education level, marital status, working status, morbidities) and history of regular health check were considered as confounding variables and were entered in the analysis last. Nagelkerke $R^2$ was used as an estimate of the variance explained by the model. The magnitude of difference in Nagelkerke $R^2$ of the hierarchical model allowed us to compare the importance of these factors sequentially. The complimentary log-log link function was used in the analysis. This link function was used because the pattern of outcome category skewed towards the higher. Model fitness was assessed using the Deviance goodness-of-fit measures [24]. The test of Parallel lines was used to assess the proportional odds assumption. Multicollinearity of the independent variables was examined using variance inflation factor (VIF). The VIFs for independent variables in the models were < 10, which indicated no multi-collinearity existed between these variables. The summary results of the presumptions checked and model adequacy using Akaike Information Criterion (AIC) were shown in S1 Table. All final models have the lowest AIC values, indicating these were the optimal models.

## Ethical issues

Participation in the survey was voluntary. Written consent was obtained from each participant. This study has obtained ethics approval from the Medical Ethics Committee, University of Malaya Medical Centre (20145–274).

## Results

There were 397 participants included in this analysis; 60% (n = 237) were women. The mean age was 48 years (SD11.5) for men and 46.5 years (SD10.8) for women. The sociodemographic characteristics of the participants are shown in Table 1.

For the degree of likeliness to undergo health checks, the frequencies of scores 1, 2 and 3 (very unlikely/unlikely/not sure) were very small. Hence, these were grouped into one category. Using chi-square tests, the associations between the two variables of interest, degree of likeliness to undergo health checks and likely timeline to undergo health checks and gender were tested. There was no difference found in the proportion of men and women and their degree of likeliness and likely timeline to undergo health checks. (Table 2).

Tables 3 and 4 show the results of the hierarchical ordinal regression analysis between the factors and the degree of likeliness and likely timeline to undergo health checks respectively

**Table 1. The sociodemographic characteristic of men and women.**

| Sociodemographic characteristics | Men | Women | p-value [*] |
|---|---|---|---|
| | (N = 160) | (N = 237) | |
| | n (%) | n (%) | |
| **Age group** | | | 0.250 |
| 30–39 | 46 (28.7) | 80 (33.8) | |
| 40–49 | 43 (26.9) | 86 (36.3) | |
| 50–59 | 41 (25.6) | 37 (15.6) | |
| ≥60 | 30 (18.8) | 34 (14.3) | |
| **Ethnicity** | | | 0.346 |
| Malay | 93 (58.1) | 118 (49.8) | |
| Chinese | 53 (33.1) | 95 (40.1) | |
| Indian | 10 (6.3) | 14 (5.9) | |
| Others | 4 (2.5) | 10 (4.2) | |
| **Education level** | | | 0.875 |
| Primary | 7 (4.4) | 8 (3.4) | |
| Secondary | 72 (45.0) | 107 (45.1) | |
| Tertiary | 81 (50.6) | 122 (51.5) | |
| **Marital status** | | | 0.523 |
| Married | 139 (86.9) | 193 (81.4) | |
| **Working status** | | | 0.001 |
| Working | 128 (80.0) | 150 (63.3) | |
| **Morbidities** | | | |
| High cholesterol | 31 (19.5) | 40 (16.9) | 0.505 |
| High blood pressure | 39 (24.5) | 34 (14.3) | 0.012 |
| Diabetes mellitus | 21 (13.2) | 19 (8.0) | 0.125 |
| **Past history of regular health checks** | 87 (54.4) | 133 (56.1) | 0.758 |

[*]chi-square test

among men and women, controlling for age group, ethnicity, education level, marital status, working status, morbidities and past history of regular health checks. Based on the Nagelkerke $R^2$ values, for both men and women, the internal factors contributed more to the differences in both outcome variables. For the men, the degree of likeliness to undergo CVD health checks

**Table 2. Intention to undergo CVD health checks among men and women.**

| Intention to undergo CVD health checks | Men | Women | p-value |
|---|---|---|---|
| | (N = 160) | (N = 237) | |
| | n (%) | n (%) | |
| **Degree of likeliness to undergo health checks** | | | 0.250 |
| not sure/unlikely/very unlikely | 21 (13.1) | 46 (19.4) | |
| Likely | 72 (45.0) | 102 (43.0) | |
| very likely | 67 (41.9) | 89 (37.6) | |
| **Likely timeline to undergo health checks** | | | 0.154 |
| Not sure/not likely to go | 21 (13.1) | 46 (19.4) | |
| Likely within 1 year | 28 (17.5) | 53 (22.4) | |
| Likely within 6 months | 38 (23.8) | 46 (19.4) | |
| Likely within 3 months | 73 (45.6) | 92 (38.8) | |

**Table 3. Determinants of individuals' degree of likeliness to undergo CVD health check.**

| Determinants | | Degree of likeliness to undergo CVD health checks | | | | | | | |
|---|---|---|---|---|---|---|---|---|---|
| | | Men | | | | Women | | | |
| | | β (95%CI) | SE | p-value | Pseudo R-square (Nagelkerke) | β (95%CI) | SE | p-value | Pseudo R-square (Nagelkerke) |
| Internal factors | Believe that the disease course can be changed for better outcomes | 0.032 (-0.636 to 0.699) | 0.341 | 0.926 | 31.6%† | 0.022 (-0.427 to 0.471) | 0.229 | 0.924 | 18.9%† |
| | Perceived self at risk of CVD | -0.073 (-0.549 to 0.403) | 0.243 | 0.765 | | -0.064 (-0.353 to 0.225) | 0.147 | 0.664 | |
| | Preferred method for CVD prevention | -0.082 (-0.467 to 0.303) | 0.197 | 0.676 | | 0.017 (-0.220 to 0.254) | 0.121 | 0.888 | |
| | Perceived benefits of health checks | 0.740 (-0.047 to 1.528) | 0.402 | 0.065 | | 0.183 (-0.302 to 0.668) | 0.247 | 0.460 | |
| | Perceived drawbacks of health checks | **-1.093 (-1.592 to -0.594)** | **0.255** | **<0.001***  | | -0.114 (-0.537 to 0.310) | 0.216 | 0.599 | |
| | Readiness to know the result of health checks | 0.272 (-0.645 to 1.190) | 0.468 | 0.561 | | 0.180 (-0.298 to 0.658) | 0.244 | 0.460 | |
| | Readiness to handle the outcomes following health checks | 0.275 (-0.512 to 1.062) | 0.402 | 0.494 | | **0.575 (0.063 to 1.087)** | **0.261** | **0.028***  | |
| External factors | External barriers | -0.165 (-0.790 to 0.461) | 0.319 | 0.606 | 1.2%‡ | **-0.440 (-0.869 to -0.011)** | **0.219** | **0.044***  | 3.1%‡ |
| | Influenced by significant others | -0.098 (-0.662 to 0.465) | 0.288 | 0.733 | | **0.406 (0.013 to 0.800)** | **0.201** | **0.043***  | |

β: Estimates of regression coefficient; SE: Standard error; CI: Confidence interval

*p < 0.05

†Pseudo R-square from first block of internal factors in hierarchical model

‡additional Pseudo R-square after second block of external factors included in the model

Controlled for age group, ethnicity, education level, marital status, working status, morbidities and past history of regular health checks

was negatively associated with perceived drawbacks of health checks (P< 0.001). For the women, the degree of likeliness to undergo CVD health checks was positively associated with readiness to handle the outcomes following health checks (P = 0.028), 'influenced by significant others' (P = 0.043) and negatively with external barriers (P = 0.044). For likely timeline to undergo CVD health checks for the men, there was again a negative association with perceived drawbacks of health checks (P = 0.018). For the women, the likely timeline to undergo CVD health checks was positively associated with readiness to handle the outcomes following health checks (P = 0.009).

## Discussion

We found that for both genders, internal factors explained a greater proportion of the variations in the intention to undergo CVD health checks compared to external factors. Significant factor associated with intention to undergo CVD health checks for men was an internal factor, perceived drawbacks of the health checks. For women, significant factors associated with intention to undergo CVD health checks included both internal factor, individuals' readiness to handle the outcomes following CVD health checks, and external factors, external barriers and influenced by significant others.

The majority of the participants in both genders had indicated similar high rates (86.9% in men and 80.6% in women) of intention to undergo CVD health checks. This reflected they were keen in CVD preventive care. This was in contrast with the general perception that men seemed uninterested in their health [25–28]. However, reviews on men's health seeking

**Table 4. Determinants of individuals' likely timeline to undergo CVD health checks.**

| Determinants | | Likely timeline to undergo CVD health checks | | | | | | | |
|---|---|---|---|---|---|---|---|---|---|
| | | Men | | | | Women | | | |
| | | β (95%CI) | SE | p-value | Pseudo R-square (Nagelkerke) | β (95%CI) | SE | p-value | Pseudo R-square (Nagelkerke) |
| Internal factors | Believe that the disease course can be changed for better outcomes | 0.351 (-0.204 to 0.906) | 0.283 | 0.216 | 9.6%† | -0.109 (-0.538 to 0.320) | 0.219 | 0.617 | 22.1%† |
| | Perceived self at risk of CVD | 0.258 (-0.131 to 0.647) | 0.198 | 0.193 | | 0.196 (-0.074 to 0.466) | 0.138 | 0.155 | |
| | Preferred method for CVD prevention | 0.078 (-0.246 to 0.402) | 0.165 | 0.636 | | -0.093 (-0.325 to 0.140) | 0.119 | 0.436 | |
| | Perceived benefits of health checks | 0.086 (-0.582 to 0.754) | 0.341 | 0.801 | | 0.234 (-0.221 to 0.689) | 0.232 | 0.314 | |
| | Perceived drawbacks of health checks | **-0.533 (-0.975 to -0.091)** | **0.225** | **0.018***| | 0.033 (-0.375 to 0.441) | 0.208 | 0.873 | |
| | Readiness to know the result of health checks | -0.433 (-1.204 to 0.337) | 0.393 | 0.271 | | 0.365 (-0.089 to 0.818) | 0.231 | 0.115 | |
| | Readiness to handle the outcomes following health checks | 0.315 (-0.334 to 0.965) | 0.331 | 0.341 | | **0.645 (0.162 to 1.128)** | **0.246** | **0.009***| |
| External factors | External barriers | 0.021 (-0.542 to 0.584) | 0.287 | 0.942 | 1.8%‡ | -0.396 (-0.804 to 0.011) | 0.208 | 0.057 | 4.2%‡ |
| | Influence by significant others | 0.061 (-0.391 to 0.514) | 0.231 | 0.790 | | 0.263 (-0.119 to 0.644) | 0.195 | 0.177 | |

β: Estimates of regression coefficient; SE: Standard error; CI: Confidence interval

*p < 0.05

†Pseudo R-square from first block of internal factors in hierarchical model

‡additional Pseudo R-square after second block of external factors included in the model

Controlled for age group, ethnicity, education level, marital status, working status, morbidities and past history of regular health checks

behaviour had noted inconsistent findings, which were attributed to differences in men's socioeconomic status rather than mere gender differences. Men with higher socio-economic status were shown to attend health checks as frequently as women [18,29]. Middle and older age men were reported to have similar interest in cardiovascular health checks compared to women [30]. Thus, men's health seeking is contextual and varied [31–33]. Our study population belonged mainly to those from middle to higher socio-economic status, with half of the participants having attained tertiary education. Furthermore, Asian culture might have impacted on health seeking behaviour, as Asian men was noted to present early to hospital following chest pain [34].

Although we showed no difference between gender in their intention to undergo CVD health checks, the factors influencing their intentions differed, with a strong influence of internal factors. Many studies that measured the impact of gender and factors on undergoing health checks focused mainly on sociodemographic factors [30,35,36] rather than a comprehensive evaluation of internal and external factors. Previous study had demonstrated a small but significant relationship between age, gender, social factors and health checks intention and behaviour, with variance explained by only less than 4% [37]. Other studies have explored similar internal and external factors, but many were limited by reporting significant factors, without quantifying the magnitude of impact on health check behaviour [38–40]. Armitage et al had examined and quantified the impact of sociodemographic and internal factors on intention to health checks, and found the variance explained were 3% and 16% respectively [37]. Quantifying the impact of external factors was not attempted [37]. We extended our study beyond

sociodemographic factors to quantify the impact of internal and external factors. The internal factors referred to personal and psychological factors such as attitude, perception and preparedness for CVD health checks, which were noted in our earlier exploratory study [23]. In this study, we found external factors had much less impact on the intention to undergo CVD health checks, with only a small increase in variance explained compared to those contributed by internal factors. Our results were consistent with the findings that interventions conducted to improve uptake of the CVD health checks focusing on external factors, such as targeting health care providers (significant others) and provide financial incentives (external barriers) to individuals, did not produce satisfactory impact [10,41]. Our local data from the SOCSO CVD health checks programme had interventions targeting external resources (giving free voucher and easy accessibility) and the result was not satisfactory [15]. A meta-analysis on determinants to predict health screening attendance has emphasised the importance of attitude as the factor predicting intention to attend screening [42]. We postulated CVD health checks behaviour was related more to psychological and personal factors and socio-demographic factors were background factors [43]. Therefore, public health intervention focusing on internal factors may result in better outcomes than on external factors.

External factors were not found to be significantly associated with health check intention in men but were significant factors in women. The external factor of particular interest was influence by significant others. Men tended to downplay minor health issues while women tended to discuss and share their experiences and knowledge with their peers, and usually sought spouses or family members' opinion before enrolling in treatment and preventive health care [32,44,45]. Women focused more on close relationship in self-construal compared to men [45]. These supported our findings that having significant others is a factor that influenced women's intention to undergo CVD health checks. In addition, women's autonomy in making health care decision varies in different culture and social structure in developing countries [46], similar to the context where this study was conducted. Women might still rely on their spouse for financial support and transportation in seeking health checks. Thus, these external factors could influence their intention to undergo CVD health checks, albeit to a lesser extent than internal factors.

The internal factors that significantly influenced both gender's intention to CVD health checks were related to the theory of health belief model and theory of reasoned action and planned behaviour (TRA) [43,47,48]. The perceived drawbacks of health checks in men, and readiness to handle the outcomes following health checks in women, were significantly associated with individual's degree of likeliness and the likely timeline to undergo CVD health checks, which was similar to the concept of self-efficacy in TRA. We did not find any studies that examined these factors separately between men and women. For men, perceived drawback refers to a sense of threat to health image resulting from detection of risk factors following health checks [23]. These threats include a need to change his lifestyle, potential exposure of health status to insurance, employer, spouse and friends. This is consistent with men's help seeking behaviour, where exposure of health status is considered a threat to their masculine image [26,32]. For women, readiness to handle the outcomes following health checks refers to individual's mental preparedness to deal with outcomes resulting from detection of risk factors following health checks [23]. This mental preparedness included readiness to take medication, adjustment of lifestyle and bearing the cost of subsequent treatment. Future health education to encourage CVD health checks needs to take into account these differences between genders to promote success of the programme.

## Strength and limitations

The strength of this study was the recruitment from community, which reflected public's perception regarding significant factors that influenced men and women to undergo CVD health

checks. The CVD health check questionnaire used was developed based on findings of an earlier qualitative study that provided an understanding of individuals' intention to undergo CVD health checks in the local context [22,23].

This study is limited by convenience sampling and the exclusion of public who could not speak and read Malay.Thus it needs to be interpreted with caution. However, there was only one participant who was excluded due to language issue and we had recruited the main ethnic groups in Malaysia (Malay, Chinese and Indian). We were unable to determine the cause and effect relationship due to the cross-sectional study design. Several biases could have occurred due to self-report data such as the socially desirable effect and recall bias from participants. We acknowledged measurement of participants' actual participation in health checks would be the desired outcome but due to resource constraints, we used intention to undergo health checks as a surrogate for action in this preliminary work. Future study can examine the effectiveness of interventions targeting these factors separately for men and women.

## Conclusion

Internal factors were significant factors associated with CVD health checks for both men and women, although external factors were significant for women too. However, there was no gender difference in the degree of likeliness and timeline to undergo health checks. Both genders had keen intention to undergo CVD health checks. Future Interventions to improve health check participation need to focus on internal factors and not limited to the commonly targeted external factors. For men, there is a need to address the perceived drawbacks of health checks to improve participation in preventive care. For women, improving their self-confidence in handling CVD health checks outcomes seemed to be of greater importance.

## Supporting information

**S1 File. The "determinants of intention to undergo CVD health checks" questionnaire.**
(PDF)

**S1 Table. Summary results of presumption checked.**
(PDF)

## Acknowledgments

We would like to thank all the participants for taking part in this study.

## Author Contributions

**Conceptualization:** Ai Theng Cheong, Seng Fah Tong, Karuthan Chinna, Ee Ming Khoo, Su May Liew.

**Data curation:** Ai Theng Cheong.

**Formal analysis:** Ai Theng Cheong, Seng Fah Tong, Karuthan Chinna.

**Funding acquisition:** Ee Ming Khoo.

**Methodology:** Ai Theng Cheong, Seng Fah Tong, Karuthan Chinna, Ee Ming Khoo, Su May Liew.

**Project administration:** Ai Theng Cheong.

**Resources:** Ee Ming Khoo.

**Supervision:** Karuthan Chinna, Ee Ming Khoo, Su May Liew.

**Validation:** Seng Fah Tong, Karuthan Chinna, Ee Ming Khoo, Su May Liew.

**Writing – original draft:** Ai Theng Cheong, Seng Fah Tong.

**Writing – review & editing:** Ai Theng Cheong, Seng Fah Tong, Karuthan Chinna, Ee Ming Khoo, Su May Liew.

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
