## [Decision Letter · Decision Letter 0]

29 Jul 2020

PONE-D-20-17378

Gender differences in factors influencing intention to undergo cardiovascular disease health checks: a cross-sectional survey

PLOS ONE

Dear Dr. Cheong,

Thank you for submitting your manuscript to PLOS ONE. After careful consideration, we feel that it has merit but does not fully meet PLOS ONE’s publication criteria as it currently stands. Therefore, we invite you to submit a revised version of the manuscript that addresses the points raised during the review process.

We look forward to receiving your revised manuscript.

Kind regards,

Amir H. Pakpour, Ph.D.

Academic Editor

PLOS ONE

Journal Requirements:

2.Please refer to any sample size calculations performed prior to participant recruitment. If these were not performed please justify the reasons. Please refer to our statistical reporting guidelines for assistance (https://journals.plos.org/plosone/s/submission-guidelines.#loc-statistical-reporting).

Reviewers' comments:

Reviewer's Responses to Questions

**Comments to the Author**

1. Is the manuscript technically sound, and do the data support the conclusions?

Reviewer #1: No

Reviewer #2: Partly

2. Has the statistical analysis been performed appropriately and rigorously? 

Reviewer #1: No

Reviewer #2: No

3. Have the authors made all data underlying the findings in their manuscript fully available?

Reviewer #1: No

Reviewer #2: Yes

4. Is the manuscript presented in an intelligible fashion and written in standard English?

Reviewer #1: Yes

Reviewer #2: Yes

5. Review Comments to the Author

Reviewer #1: Dear author(s),

Many thanks for all efforts. Gender differences is one of the insurmountable issues not often dealt with. sexism is always questionable to specialists. your manuscript is overall reasonable and appreciable. However, there are some suggestion below:

1.In Abstract-Results part, please report coefficients with p-value and/or confidence interval.

2.In Abstract-Methods part, please express further information about statistical analysis and present the method of inference.

3.Abstract-Background is too short. please write a bit more about the previous studies before giving the aim of the study.

4.In Abstract- Results part, please follow the usual frame of writing integers, particularly,indexes. I mean SD. please write confidence intervals and p-value with them.

5.In method part, why did not explicate the type of sampling? How did you achieve the number of sub-sample?Is there any way to reach the original data?

6.why did you use hierarchical regression?Were the presumptions checked?

7.How did you apply Pearson correlation while you did use ordinal regression and proportional odds?

8.If you would like to apply log-log function?you should use AIC or BIC for model adequacy checking?

9.why was Pseudo-R2 used?

10.why did you cite some references in your result part???

11.why does not exist any plot or related path-diagram?

Reviewer #2: The study entitled “Gender differences in factors influencing intention to undergo cardiovascular disease health checks: a cross-sectional survey” used a cross-sectional design on a conveniences ample to assess the factors influencing individuals’ intention to undergo cardiovascular disease (CVD) health checks. Below please see my comments.

1. The meanings of internal factor and external factor are unclear. The authors should provide clear definition for internal factor and external factor in both Abstract and the main text.

2. The regression model should include one more block to control the impacts of demographics. That is, age, educational level, marital status, working status, and morbidities should be controlled in all the regression models.

3. The comparisons between males and females are not based on inferential test. The authors should use inferential test to make the comparisons. Please refer to the following papers for more information.

Wong, H. Y., Mo, H. Y., Potenza, M. N., Chan, M. N. M., Lau, W. M., Chui, T. K., Pakpour, A. H., Lin, C.-Y. (2020). Relationships Between Severity of Internet Gaming Disorder, Severity of Problematic Social Media Use, Sleep Quality and Psychological Distress. International Journal of Environmental Research and Public Health, 17, 1879.

Ou, H.-t., Su, C.-T., Luh, W.-M., & Lin, C.-Y. (2017). Knowing is half the battle: The association between leisure-time physical activity and quality of life among four groups with different self-perceived health status in Taiwan. Applied Research in Quality of Life, 12(4), 799-812.

4. Some languages are not academic (e.g., “This was a subgroup analysis of a bigger study”), I would recommend the authors finding assistance from a professional English editor.

5. The limitations of cross-sectional design and self-reports should be acknowledged. Also, the study did not assess the actual behavior of CVD health checks among these participants. Therefore, it is hard to conclude whether the significant factors are really useful in promoting CVD health checks.

6. The footnote of Table 3 mentions “SE: standard error”. However, Table 3 does not report SE.

6. PLOS authors have the option to publish the peer review history of their article (what does this mean?). If published, this will include your full peer review and any attached files.

Reviewer #1: No

Reviewer #2: No

---

## [Author Response · Author response to Decision Letter 0]

9 Sep 2020

Answer to reviewers’ comments

Journal Requirements:

We have checked and complied to the requirements.

2.Please refer to any sample size calculations performed prior to participant recruitment. If these were not performed please justify the reasons. Please refer to our statistical reporting guidelines for assistance (https://journals.plos.org/plosone/s/submission-guidelines.#loc-statistical-reporting).

Sample size calculation was added. (study design and data collection, page 6-7, line 161-168)

For the regression model, the sample size needed was calculated based on the work of Peduzzi et al [23] using the following formula: 

 N=10k/P

Where k was the number of independent variables (9 factors were examined in this study) and P was the proportions of intention to undergo CVD health check. The P was estimated from the intention to undergo for CVD health check reported as 83.7% [20]. Thus, the estimated sample size required was 108 for men and women respectively. The larger study recruited 397 participants, providing adequate power for this analysis. 

Reviewer #1: Dear author(s),

Many thanks for all efforts. Gender differences is one of the insurmountable issues not often dealt with. sexism is always questionable to specialists. your manuscript is overall reasonable and appreciable. However, there are some suggestion below:

Thank you for the comments and suggestions.

1.In Abstract-Results part, please report coefficients with p-value and/or confidence interval. 

Revised. (abstract, results, page 2-3, line 47-59)

2.In Abstract-Methods part, please express further information about statistical analysis and present the method of inference.

Revised. (abstract, method, page 2, line 42-44)

3.Abstract-Background is too short. please write a bit more about the previous studies before giving the aim of the study. 

Revision done. Due to word count limitation in abstract, we choose to focus on the methods and results.

(abstract, background, page 2, line 31-34)

Undergo a health check for cardiovascular disease (CVD) is an important strategy to improve cardiovascular (CV) health. Men are reported to be less likely to undergo cardiovascular disease (CVD) health check than women. Gender difference could be one of the factors influencing health seeking behaviour of men and women.

4.In Abstract- Results part, please follow the usual frame of writing integers, particularly,indexes. I mean SD. please write confidence intervals and p-value with them. 

Revised. (abstract, results, page 2-3, line 47-59)

5.In method part, why did not explicate the type of sampling? 

We have added the information. (method, page 5, line 136-138)

This was a cross sectional survey using mall intercept interviews [21]. Malaysians aged >30 years who attended a hypermarket were invited to participate in the survey using convenient sampling.

How did you achieve the number of sub-sample?

As above. (study design and data collection, page 6-7, line 161-168)

For the regression model, the sample size needed was calculated based on the work of Peduzzi et al [23] using the following formula: 

 N=10k/P

Where k was the number of independent variables (9 factors were examined in this study) and P was the proportions of intention to undergo CVD health check. The P was estimated from the intention to undergo for CVD health check reported as 83.7% [20]. Thus, the estimated sample size required was 108 for men and women respectively. The larger study recruited 397 participants, providing adequate power for this analysis.

Is there any way to reach the original data?

The original data can be requested from the author if needed. 

6.why did you use hierarchical regression?

Clarified. (data analysis, page 7-8, line 192-194)

This hierarchical ordinal regression was chosen because we would like to know in addition to the internal factors, the effect of external factors on participants’ intention to undergo CVD health checks. 

 Were the presumptions checked?

Revised to add clarity. (data analysis, page 8, line 203-210)

Model fitness was assessed using the Deviance goodness-of-fit measures [24]. The test of Parallel lines was used to assess the proportional odds assumption. Multicollinearity of the independent variables was examined using variance inflation factor (VIF). The VIFs for independent variables in the models were < 10, which indicated no multicollinearity existed between these variables. The summary results of the presumptions checked and model adequacy using Akaike Information Criterion (AIC) were shown in S1 Table. All final models have the lowest AIC values, indicating these were the optimal models.

7.How did you apply Pearson correlation while you did use ordinal regression and proportional odds?

We used Pearson correlation to check collinearity between the 9 factors. We have now removed this and used VIF to check for multicollinearity of the independent variables. (data analysis, page 8, line 205-208)

8.If you would like to apply log-log function? you should use AIC or BIC for model adequacy checking?

Thank you for the suggestions.

We have revised as suggested (data analysis, page 8, line 208-210) and included the AIC results in S1 Table. 

9.why was Pseudo-R2 used?

We would like to know in addition to the internal factors, the effect of external factors by comparing the pseudo-R2.

Revised as follows (data analysis, page 8, line 199-201).

Nagelkerke R2 was used as an estimate of the variance explained by the model. The magnitude of difference in Nagelkerke R2 of the hierarchical model allowed us to compare the importance of these factors sequentially. 

10.why did you cite some references in your result part???

We have rechecked the result section and did not cite any references in this section.

11.why does not exist any plot or related path-diagram? 

We did not test any path effects.

Reviewer #2: The study entitled “Gender differences in factors influencing intention to undergo cardiovascular disease health checks: a cross-sectional survey” used a cross-sectional design on a conveniences sample to assess the factors influencing individuals’ intention to undergo cardiovascular disease (CVD) health checks. Below please see my comments.

Thank you for the comments and suggestions.

1. The meanings of internal factor and external factor are unclear. The authors should provide clear definition for internal factor and external factor in both Abstract and the main text.

We have added the definitions for internal and external factors in the abstract (abstract, method, page 2, line 40-42) and study design and data collection section (page 6, line 146-147). 

Seven were internal factors that were related to individuals’ attitude, perception and preparedness for CVD health checks and two external factors that were related to external resources [23].

2. The regression model should include one more block to control the impacts of demographics. That is, age, educational level, marital status, working status, and morbidities should be controlled in all the regression models.

Revised (data analysis, page 8, line 196-199)

…, the internal factors were included in the first block of the regression followed by the external factors. The demographics (age group, education level, marital status, working status, morbidities) and history of regular health check were considered as confounding variables and were entered in the analysis last.

The results were presented in table 3 and 4. 

3. The comparisons between males and females are not based on inferential test. The authors should use inferential test to make the comparisons. Please refer to the following papers for more information.

Wong, H. Y., Mo, H. Y., Potenza, M. N., Chan, M. N. M., Lau, W. M., Chui, T. K., Pakpour, A. H., Lin, C.-Y. (2020). Relationships Between Severity of Internet Gaming Disorder, Severity of Problematic Social Media Use, Sleep Quality and Psychological Distress. International Journal of Environmental Research and Public Health, 17, 1879.

Ou, H.-t., Su, C.-T., Luh, W.-M., & Lin, C.-Y. (2017). Knowing is half the battle: The association between leisure-time physical activity and quality of life among four groups with different self-perceived health status in Taiwan. Applied Research in Quality of Life, 12(4), 799-812.

Thank you for the suggestion.

However, our objective was to identify significant factors affecting men and women in undergoing health checks, and to compare the differences in the factors identified between men and women. 

Inferential test to compare the coefficients across men and women was not conducted because we did not intend to compare the magnitude differences between these coefficients. 

4. Some languages are not academic (e.g., “This was a subgroup analysis of a bigger study”), I would recommend the authors finding assistance from a professional English editor.

We have edited according to formal language as much as we can. 

Revision done (page 5, line 131-133)

This was a sub-analysis of a larger study that examined factors influencing individuals’ intention to undergo CVD health checks.

5. The limitations of cross-sectional design and self-reports should be acknowledged. 

Also, the study did not assess the actual behavior of CVD health checks among these participants. Therefore, it is hard to conclude whether the significant factors are really useful in promoting CVD health checks.

We have added the limitations in the section of strength and limitations (page 16, line 347-354)

We were unable to determine the cause and effect relationship due to the cross-sectional study design. Several biases could have occurred due to self-report data such as the socially desirable effect and recall bias from participants. We acknowledged measurement of participants’ actual participation in health checks would be the desired outcome but due to resource constraints, we used intention to undergo health checks as a surrogate for action in this preliminary work. Future study can examine the effectiveness of interventions targeting these factors separately for men and women.

6. The footnote of Table 3 mentions “SE: standard error”. However, Table 3 does not report SE. 

We have added SE values in Table 3 and 4.

---

## [Editor Report · Decision Letter 1]

11 Sep 2020

Gender differences in factors influencing intention to undergo cardiovascular disease health checks: a cross-sectional survey

PONE-D-20-17378R1

Dear Dr. Cheong,

We’re pleased to inform you that your manuscript has been judged scientifically suitable for publication and will be formally accepted for publication once it meets all outstanding technical requirements.

Kind regards,

Amir H. Pakpour, Ph.D.

Academic Editor

PLOS ONE
---

## [Editor Report · Acceptance letter]

16 Sep 2020

PONE-D-20-17378R1 

Gender differences in factors influencing intention to undergo cardiovascular disease health checks: a cross-sectional survey 

Dear Dr. Cheong:

I'm pleased to inform you that your manuscript has been deemed suitable for publication in PLOS ONE. Congratulations! Your manuscript is now with our production department. 

Kind regards, 

on behalf of

Dr. Amir H. Pakpour 

Academic Editor

PLOS ONE